# A Frequency-Reconfigurable Filtenna for GSM, 4G-LTE, ISM, and 5G-Sub 6 GHz Band Applications

**DOI:** 10.3390/s22155558

**Published:** 2022-07-25

**Authors:** Wahaj Abbas Awan, Niamat Hussain, Sunggoo Kim, Nam Kim

**Affiliations:** 1Department of Information and Communication Engineering, Chungbuk National University, Cheongju 28644, Korea; wahajabbasawan@chungbuk.ac.kr (W.A.A.); ksq39@naver.com (S.K.); 2Department of Smart Device Engineering, Sejong University, Seoul 05006, Korea; niamathussain@sejong.ac.kr

**Keywords:** filtenna, frequency reconfigurable, flexible antenna, GSM, LTE, ISM, 5G-sub 6 GHz

## Abstract

This paper presents the design and realization of a flexible and frequency-reconfigurable antenna with harmonic suppression for multiple wireless applications. The antenna structure is derived from a quarter-wave monopole by etching slots. Afterward, the high-order unwanted harmonics are eliminated by adding a filtering stub to the feedline to avoid signal interference. Lastly, frequency reconfigurability is achieved using pin diodes by connecting and disconnecting the stubs and the rectangular patch. The antenna is fabricated on the commercially available thin (0.254 mm) conformal substrate of Rogers RT5880. The proposed antenna resonates (|S_11_| < –10 dB) at five different reconfigurable bands of 3.5 GHz (3.17–3.82 GHz), 2.45 GHz (2.27–2.64 GHz), 2.1 GHz (2.02–2.29 GHz), 1.9 GHz (1.81–2.05 GHz), and 1.8 GHz (1.66–1.93 GHz), which are globally used for 5G sub-6 GHz in industrial, medical, and scientific (ISM) bands, 4G long-term evolution (LTE) bands, and global system for mobile communication (GSM) bands. The simulated and measured results show that the antenna offers excellent performance in terms of good impedance matching with controllable resonant bands, high gain (>2 dBi), stable radiation patterns, and efficiency (>87%). Moreover, the conformal analysis shows that the antenna retains its performance both in flat and bending conditions, making it suitable for flexible electronics. In addition, the antenna is compared with the state-of-the-art works for similar applications to show its potential for the targeted band spectrums.

## 1. Introduction

The unwanted higher-order harmonics interfere with the signals of the nearby Radio Frequency (RF) circuitry within a wireless electronic system, reducing the wireless system’s overall performance. These harmonics are generated by various electric components [1]. In compact electronic devices, antenna elements also create harmonics at higher frequencies due to the small physical size of the radiating structure [2]. Therefore, additional filters are required to mitigate electromagnetic interference with the signals of another RF circuitry in the nearby region [3]. However, filters increase insertion loss and the device’s size [4]. This has led to the realization of antennas with the inbuilt filters (filtennas) using cutting slits/slots, band rejection cells, defected ground structure, and electromagnetic bandgap structures to make the system more compact and efficient [5,6,7,8,9].

In addition, stretchable electronics have emerged in all fields of life, from personal/medical/industrial usages to sensing devices. It has resulted in a drastic demand for conformal antennas for effective communication [10]. Researchers have exploited patch and monopole antennas for conformal applications due to their simple design. However, they suffer from large dimensions, limiting their usage for compact devices [11,12]. The antenna size has been reduced by employing various open-ended or short-ended stubs and slots into radiating structures. Nevertheless, it increases the structural complexity, which results in fabrication issues [13]. Recent work shows the utilization of several types of elastic materials for the design of conformal antennas that are not limited to textile materials, including jeans, conductive textile–polymer composites, and polydimethylsiloxane polymer [14,15,16].

The antenna should operate on a reconfigurable multiband to satisfy the end-user requirements. For this purpose, several studies have been conducted on bendable and frequency-reconfigurable designs [17,18,19,20,21,22]. Various kinds of multiband antennas for heterogenous flexible applications were presented in [17,18,19,20]. These works offer relatively simple geometrical configurations, along with broadband at all resonances. However, these designs suffer from the physically large size, which limits their use for compact-size devices. Contrary to this, compact-size antennas were presented in [21,22]. The antennas offer a moderate gain with quad-band and tri-band operations, respectively. Still, none of the above-mentioned antennas have provided any solution for harmonic suppression.

Significant efforts have been made in the literature to design antennas capable of harmonic suppression [23,24,25,26,27,28]. The antennas operating at the 2.45 GHz band offer the advantage of high-order harmonics suppression up to 8 GHz [23,24,25,26]. Although the antennas provide high gain and broadband characteristics, they have larger dimensions, rigidness, and a single-band operation. In addition, a rigid frequency-reconfigurable antenna that has harmonic suppression has been presented for 5G sub-6 GHz applications [27]. The antenna is backed with an artificial magnetic conductor (AMC), which helps to achieve a high gain of >6 dBi, an essential requirement for 5G outdoor applications. Another interesting work was reported in [28], wherein a folded slot antenna was proposed for 1.4 GHz and 3 GHz applications. The antenna offers the advantages of frequency reconfigurability and flexibility at the cost of a large physical size.

Concluding the discussion, the reported works either have the disadvantages of rigid substrates, larger dimensions, or fixed operating frequencies. Thereby, this work presents the design and realization of an antenna that is compact in size, conformal, and has on-demand frequency-switching between the five different bands of 1.8, 1.9, 2.1, 2.45, and 3.5 GHz, which are well-known GSM, ISM, LTE 5G sub-6 GHz bands. The antenna is designed on flexible material and retains its performance in bending conditions. Furthermore, this antenna also offers harmonic suppression up to 10 GHz to avoid potential electromagnetic interference with other systems.

## 2. Antenna Design Methodology

### 2.1. Antenna Geometry

The geometrical configuration of the proposed antenna is illustrated in Figure 1. The antenna geometry was etched on the top side of an elastic substrate, Rogers droid RT5880, having a thickness (*H*) of 0.254 mm, along with relative permittivity (*ɛ_r_*) and a loss tangent (*tan**δ*) of 2.2 and 0.009, respectively [29]. The antenna’s radiator is fed using a coplanar waveguide (CPW) feeding scheme, having virtual ground planes with dimensions *C_L_* × *C_W_*. The primary antenna consists of an arrow-shaped radiator connected to a rectangular patch, which is further connected to two inverted L-shaped serpentines through RF-pin diodes. An open-ended horizontally placed filtering stub with a length of 2 × (*C_L_* + *f*), and a width of *S_X_* is utilized to mitigate the higher-order harmonics. The backside of the antenna contains the biasing circuit to provide a controlled direct current to switch the state of the diode. The biasing pads are connected with radiating geometry using vias, whereas the inductor (*L* = 68 nH) is utilized to stop the unnecessary flow of the RF-current from the DC source, and the capacitor (*C* = 100 pF) controls the flow of the current toward the connector [30]. Figure 2 shows the equivalent model of the diode (model HPND 4005) utilized to achieve reconfigurability.

The optimized parameters of the proposed antenna are as follows: *A_L_* = 25, *A_W_* = 32, *H* = 0.254, *C_L_* = 11, *C_W_* = 5, *d* = 0.75, *f* = 1.5, *G*_1_ = 1, *P_W_* = 6.85, *S_X_* = 1, *L*_1_ = 12, *L*_2_ = 19, *L*_3_ = 7, *L*_4_ = 12, *L*_5_ = 1, *L*_6_ = 6, and *P_L_* = 5 (all units are in mm).

### 2.2. Antenna Design Methodology

The antenna design methodology of the proposed frequency-reconfigurable filtenna consists of two major steps:

1: Designing of the filtering antenna.

2: Filter antenna with frequency reconfigurability.

#### 2.2.1. Filtering Antenna

In the first step, the filtering antenna design is extracted from a CPW-fed pentagon-shaped quarter-wave monopole antenna, as depicted in Figure 3a. The effective radius (*R_eff_*) of the pentagon can be estimated by using the following relation provided in [31].
(1)$$R_{eff}=R\sqrt{1+\frac{2H}{\pi \varepsilon_{r}R}\left(ln\left(\frac{\pi R}{2H}\right)+1.7726\right)}$$
where *H* is the substrate thickness, *R* is the radius of the monopole, *ɛ_r_* is the relative permittivity of the substrate, and *π* is a constant with a value of $\frac{22}{7}$.

The effective radius of the pentagon estimates the resonating frequency (*f_r_*) using the following relation:(2)$$f_{r}=\frac{1.8412\times c}{4\pi \;R_{eff}\sqrt{\varepsilon_{r}}}$$
where *c* represents the free-space speed of light equal to 3 × 10^8^ ms^−1^.

The resultant antenna shows the resonating frequency of 3.5 GHz having an |S_11_| < −10 dB bandwidth from 3.19–3.92 GHz, corresponding to a fractional bandwidth of 20.8% (Figure 3b). To achieve compactness in antenna size, two rectangular slots that have a length (*L_S_*) of 12 mm and a width (*W_S_*) of 4.68 mm are etched in the pentagonal-shaped radiator. These slots change the shape of the radiator from the pentagon to an inverted arrow, thus increasing the effective area of the radiator, which causes the frequency to shift toward a lower frequency [32]. The antenna is then optimized to resonate at the central frequency of 3.5 GHz, having an impedance bandwidth of 660 MHz ranging from 3.3–3.96 GHz, as depicted in Figure 3b. Furthermore, it is observed that the antenna shows an additional resonance of 8.6 GHz, having a –10 dB bandwidth of 7.3–10.2 GHz. This extra resonance is the harmonics of the fundamental resonating frequency, and it may cause an unwanted interaction with the UWB communication system [23]. Thus, to overcome this unwanted interaction, there is a need to mitigate the high-order resonances.

The typical way to mitigate the harmonics from an antenna is to utilize a separate low-pass or band-stop filter [24,25,26]. However, several setbacks are associated with the usage of additional filters, which are not limited to the increase in the size of the antenna, the need for another matching circuit to match impedances of the antenna with the filter, and so on [24]. The other way is to codesign the filtering structure along with the radiating part of the antenna. Both open-ended and short-ended stubs have effectively filtered the harmonics [26]. In the present work, a rectangular stub is loaded between the radiator and the feedline without changing the overall size of the antenna, as depicted in Step-3 of Figure 3a. The dimensions of the filtering stub were optimized to achieve harmonic suppression from 5 to 10 GHz.

To further understand the effects of the filtering stub on the performance of the antenna, the current distribution and gain plot of the antenna with and without the stub is presented in Figure 4. The current distribution graphs were plotted at the high-order resonance frequency of 8.5 GHz. In the absence of the stub, the current finds an easy path to flow across the radiator, resulting in the generation of higher-mode resonance, as shown in Figure 4a. Contrary to this, when the stub is loaded into the antenna, the stub stops flowing toward the radiator and thus results in mismatched performance at higher frequencies, which eventually mitigates the higher mode, as depicted in Figure 4b. Furthermore, the antenna’s peak gain plots also verify that the stub’s filtering effect causes the decrease in the gain at higher frequencies, as shown in Figure 4c.

#### 2.2.2. Frequency-Reconfigurable Filtering Antenna

Due to the rapid evolution of the various technologies, the compact antenna must be incorporated with on-demand frequency reconfigurability to replace the usage of multiple antennas resonating at different frequencies [17,18]. Therefore, the compact filtering antenna designed in the previous stage is further utilized to develop a reconfigurable antenna using open-ended stubs, as illustrated in Figure 5a. The fundamental resonating frequency of the inserted stub can be estimated using the following relation provided in [32]:(3)$$f_{r}=\frac{c}{x_{0}L_{t}\sqrt{\varepsilon_{eff}}}$$
where *L_t_* is the total length of the stub connected to the radiating structure, and *ε_eff_* is the effective dielectric constant for *A_L_*/*H* < 1. It can be calculated using the following relation:(4)$$\varepsilon_{eff}=\frac{\varepsilon_{r}+1}{2}+\frac{\varepsilon_{r}-1}{2}\left\{{\left(1+12\frac{H}{A_{L}}\right)}^{-0.5}+0.04{\left(1-\frac{A_{L}}{H}\right)}^{2}\right\}$$

The fundamental frequency of 2.45 GHz of the radiator shifted to 2.1 GHz and 1.9 GHz utilizing two inverted L-shaped stubs with total lengths of 19 mm and 26 mm (shown in Figure 5b), respectively. To utilize the effect of the radiator and stubs separately in various combinations, two RF-Pin diodes were inserted between the radiator and the stubs for frequency reconfigurability. Moreover, due to the enormous demand for the 5G sub-6 GHz band spectrum, the resonating frequency of 3.5 GHz was also achieved by etching a slot from the radiator, which shifted the frequency towards the higher band. The final prototype of the antenna contains three-pin diodes and offers five combinational modes, Case-000, Case-100, Case-110, Case-101, and Case-111, which have resonating frequencies of 3.5 GHz, 2.45 GHz, 2.1 GHz, 1.9 GHz, and 1.8 GHz, respectively.

Figure 6 presents the |S_11_| (dB) of the proposed antenna’s |S_11_| for various states of diodes to show the frequency reconfigurability. Each switching condition has also been plotted to describe the radiation mechanism (Figure 7). For Case-000, when all diodes were in off-state, the maximum current distribution was around the lower part of the radiator, which corresponds to the radiator’s smaller electrical length, resulting in the resonance at the higher frequency band of 3.5 GHz. Alternatively, when all diodes were in on-state (Case-111), all of the radiating parts were connected to make a monopole with a larger electrical size. The surface current was distributed in the whole radiator, generating the lower resonance at 1.8 GHz. Similarly, for Case-100, the current was concentrated only on the lower and middle parts of the radiator to give resonance at 2.45 GHz. However, for the Case-110 and Case-101, the antenna operated at frequency bands of 2.1 GHz and 1.9 GHz, respectively.

## 3. Result and Discussion

### 3.1. Measurement Setup

A sample prototype of the proposed filtenna was fabricated using a standard chemical etching process to validate the design concept. Figure 8 shows the top and bottom sides of the fabricated prototype. A vector network analyzer (VNA) by ShockLine™ (model ME7868A) was utilized to measure the s-parameters. The pin diodes were connected to a 3 V battery with biasing wires. The far-field was measured in the anechoic chamber. A standard horn antenna of SGH-Series, having a peak gain of 24 dBi, was utilized inside an anechoic chamber.

### 3.2. Reflection Coefficient of the Antenna

The comparison among simulated and measured reflection coefficients of the proposed antenna for various switching cases is plotted in Figure 9. For all diodes in off-state (Case-000), the antenna resonated at 3.5 GHz, with a measured fractional bandwidth of 18.6%, ranging from 3.17–3.82 GHz. On the other hand, when diode D_1_ was kept on, while the other diode was kept in off-state (Case-100), the antenna gave a resonance at 2.45 GHz, having measured |S_11_| < –10 dB impedance bandwidth of 15.2% (2.27–2.64 GHz). Similarly, keeping D_1_ in on-state, when either D_2_ or D_3_ was kept on, and while keeping the other in off-state, then the antenna operated at 2.1 GHz and 1.9 GHz for Case-110 and Case-101, respectively. In the last case, when all the diodes were in on-state (Case-111), the antenna operated at 1.8 GHz, with a measured fractional bandwidth of 15% (1.66–1.93 GHz). Generally, an acceptable agreement between the measured and simulated results was observed for all switching states. However, the slight discrepancy between predicted and measured results is due to fabrication and measurement setup tolerances.

### 3.3. Conformal Analysis of the Filtenna

Figure 10 shows the fabricated prototype in bending conditions along circular foam with a radius of 25 mm, which is commonly used for flexible devices. The simulated and measured |S_11_| of the proposed antenna under conformal conditions is plotted in Figure 11. It can be observed that the antenna offers a strong agreement between simulated and measured results for both bending scenarios (bending along *x*- and *y*-axis). Moreover, the antenna shows nearly identical results in flat and bending scenarios, showing the proposed antenna’s performance stability for rigid and conformal applications.

### 3.4. Far-Field Parameters

For different switching states, the radiation pattern of the proposed antenna at the selected frequencies of 3.5 GHz, 2.45 GHz, 2.1 GHz, 1.9 GHz, and 1.8 GHz are shown in Figure 12, Figure 13, Figure 14, Figure 15 and Figure 16, respectively. In all cases, the antenna offers an omnidirectional radiation pattern in the principal H-plane (Φ = 0°), and monopole-like bidirectional radiation for the principal E-Plane (Φ = 90°). Likewise, when the antenna is bent along the *x*- or *y*-axis, the radiation behavior of the antenna remains identical to that of the flat case. A slight change is observed in the backward radiation, which is caused by the bending of the antenna.

The simulated and measured gain in passbands along with a simulated radiation efficiency are shown in Figure 17. For all possible operating states, the antenna offers a measured gain of more than 2 dBi in the lower resonating frequencies, while at higher frequencies it is around 3 dBi. Similarly, the antenna has a radiation efficiency of >87% for all diode switching states, depicting the operation stability of the antenna.

## 4. Comparison with State-of-the-Art Antennas

The comparison of the proposed frequency-reconfigurable flexible antenna with related works reported in the literature is summarized in Table 1. Although the antenna reported in [15] offers flexibility, it had a single operational band and a rigid structure without any harmonic suppression mechanism. On the other hand, the antennas presented in [17,18,19,20,21,22] offer flexibility and frequency reconfigurability. Yet, these designs did not show any solution to suppress high-order modes. Likewise, techniques presented in [23,24,25,26] have the advantages of filtering characteristics, but they do not provide reconfigurability conformability. Nevertheless, the reported antenna in [27] had frequency reconfigurability, harmonic suppression, bigger dimensions, and few operating bands. Lastly, a frequency-reconfigurable and flexible antenna with a relatively compact size was presented in [28]. Although the antenna offered harmonic suppression, it only covered two bands, with no information on the gain of the antenna. Thus, our design overperforms the related works by providing compact size, five modes of on-demand operations, and flexibility, making it a protentional candidate for present and future flexible electronic systems of compact size. 

## 5. Conclusions

The design and characterization of a frequency-reconfigurable flexible antenna for heterogeneous applications was presented in this paper. The antenna was derived from a conventional pentagonal-shaped quarter-wave monopole antenna. The improvement in impedance matching and size reduction was the result of truncating the corner of the radiator. In the next phase, a low bandpass stub between the feedline and the radiator stopped high-order harmonics. Finally, frequency reconfigurability was realized by utilizing the combination of serpentine structures and RF-Pin diodes. The proposed antenna shows that the five different operational bands have –10 dB impedance bandwidths of 3.17–3.82 GHz, 2.27–2.64 GHz, 2.02–2.29 GHz, 1.81–2.05 GHz, and 1.66–1.93 GHz, depending upon the state of the pin diodes. Furthermore, a stable radiation pattern, a gain of >2 dBi, and a radiation efficiency of >87% were observed at the resonating frequencies of 3.5, 2.45, 2.1, 1.9, and 1.8 GHz for case–000, case–100, case–110, case–101, and case–111, respectively. In addition, the antenna performance does not differ for flat and conformal states. Furthermore, the antenna’s comparison with state-of-the-art work shows its potential for the targeted applications operating in band spectrums allocated for 5G sub-6 GHz, 4G LTE, ISM, and GSM applications.

## Figures and Tables

**Figure 1 sensors-22-05558-f001:**
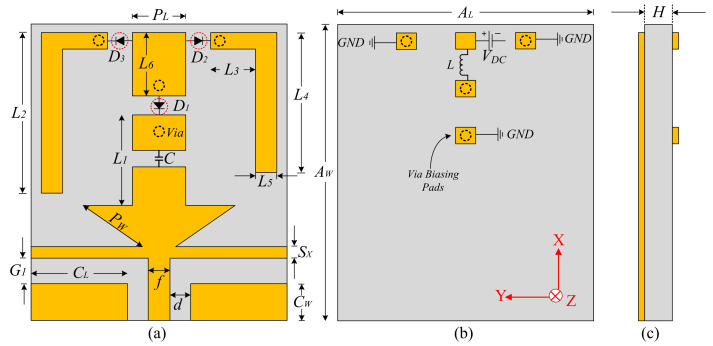
Geometrical configuration of the flexible and reconfigurable filtenna. (**a**) Top view. (**b**) Bottom view. (**c**) Side view.

**Figure 2 sensors-22-05558-f002:**
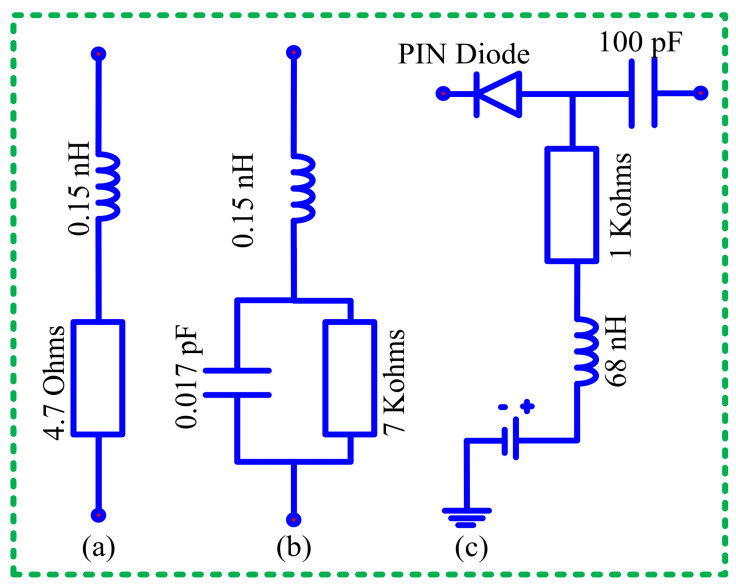
Equivalent model of diode. (**a**) On-state. (**b**) Off-state. (**c**) Biasing circuit.

**Figure 3 sensors-22-05558-f003:**
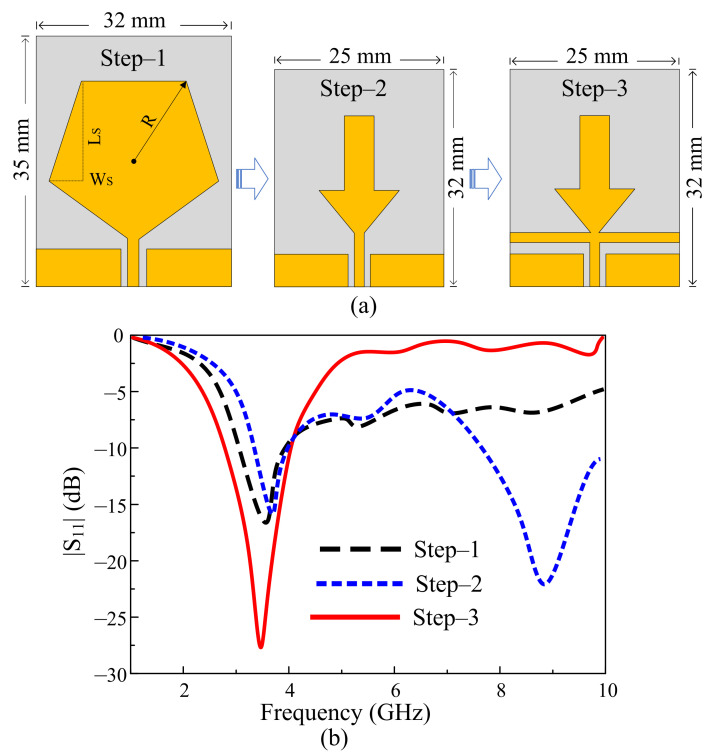
Design methodology of proposed filtenna. (**a**) Geometrical evolution. (**b**) |S_11_| for various design steps.

**Figure 4 sensors-22-05558-f004:**
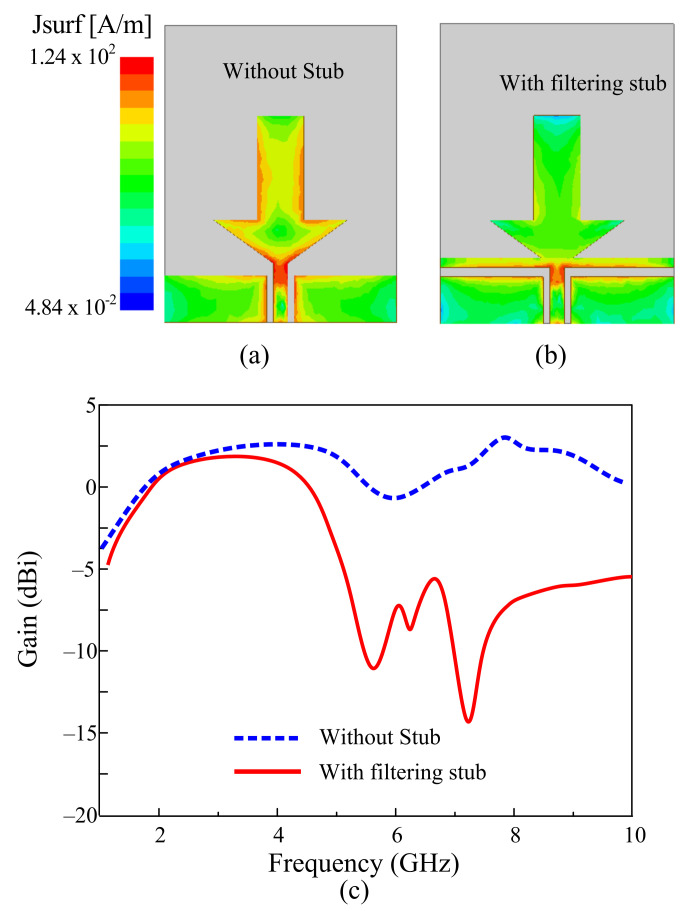
Surface current distribution of the antenna at 8.5 GHz. (**a**) Antenna without stub. (**b**) Antenna with stub. (**c**) Gain of the antenna.

**Figure 5 sensors-22-05558-f005:**
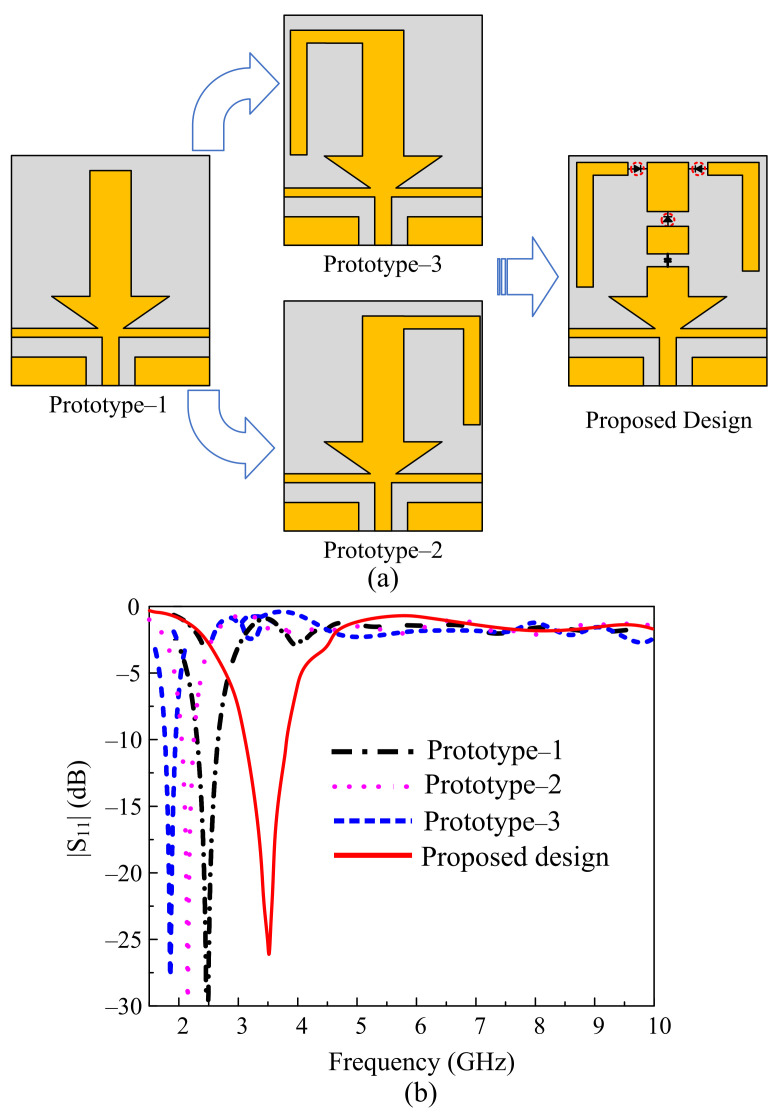
The steps involved in designing the proposed filtenna. (**a**) Geometrical configuration and (**b**) their |S_11_| responses.

**Figure 6 sensors-22-05558-f006:**
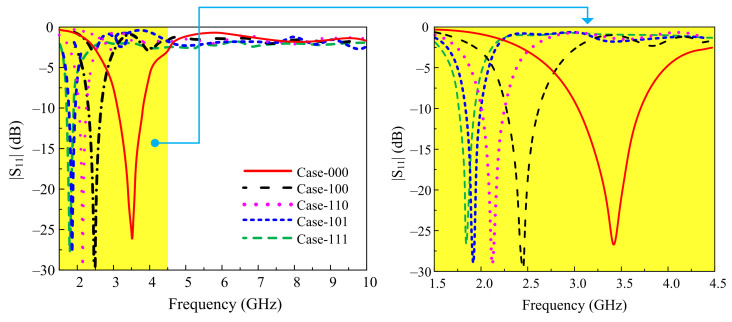
Simulated |S_11_| of proposed antenna for various switching states of diodes.

**Figure 7 sensors-22-05558-f007:**
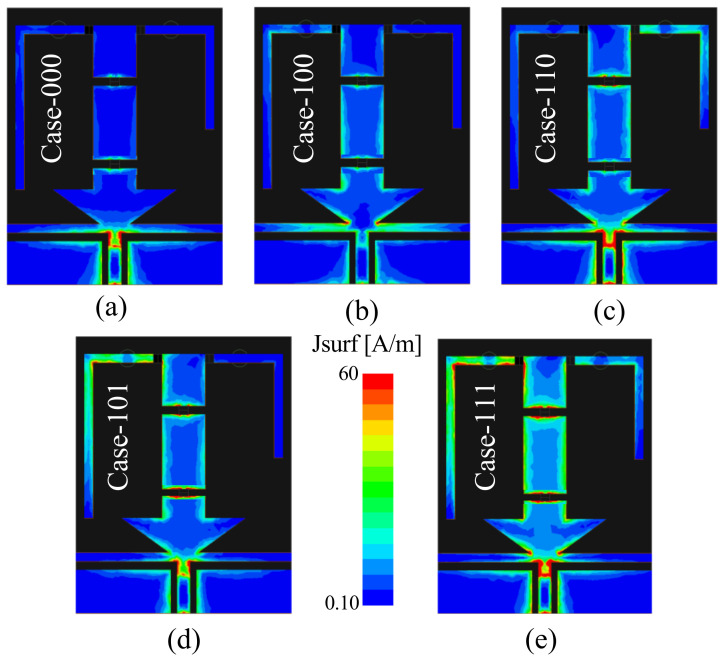
Current distribution of proposed antenna for various switch states: (**a**) Case-000, 3.5 GHz; (**b**) Case-100, 2.45 GHz; (**c**) Case-110, 2.1 GHz; (**d**) Case-101, 1.9 GHz; (**e**) Case-111, 1.8 GHz.

**Figure 8 sensors-22-05558-f008:**
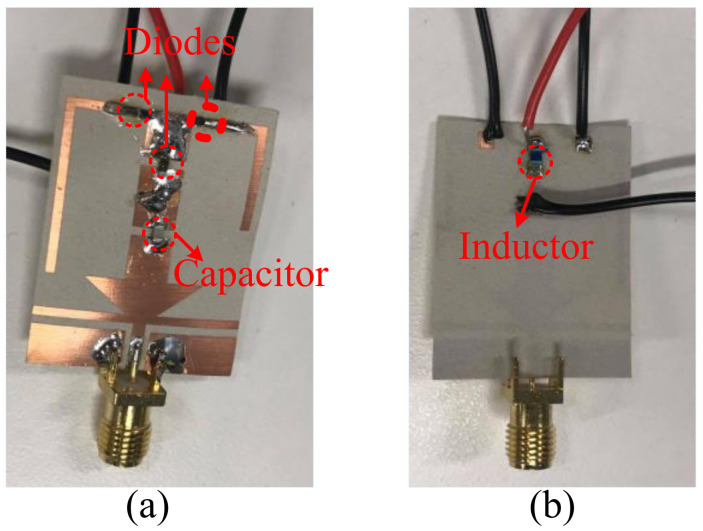
Fabricated prototype of the proposed flexible filtenna. (**a**) Top view. (**b**) Bottom view.

**Figure 9 sensors-22-05558-f009:**
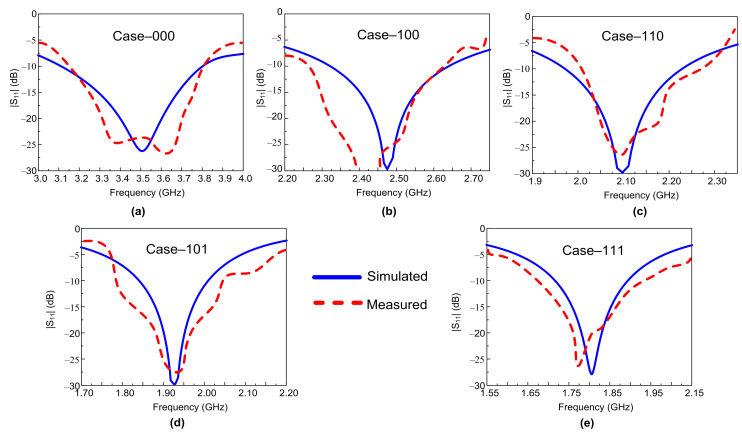
Simulated and measured |S_11_| proposed antenna for various switch states. (**a**) Case-000; (**b**) Case-100; (**c**) Case-110; (**d**) Case-101; (**e**) Case-111.

**Figure 10 sensors-22-05558-f010:**
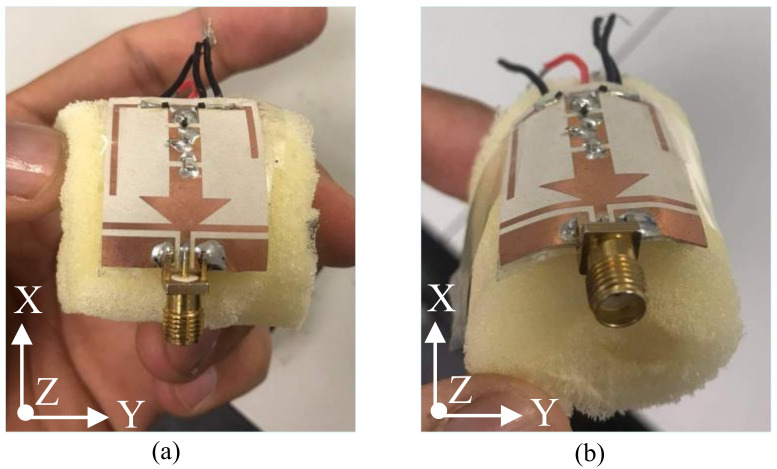
Bending conditions of the proposed filtenna along (**a**) *x*-axis and (**b**) *y*-axis.

**Figure 11 sensors-22-05558-f011:**
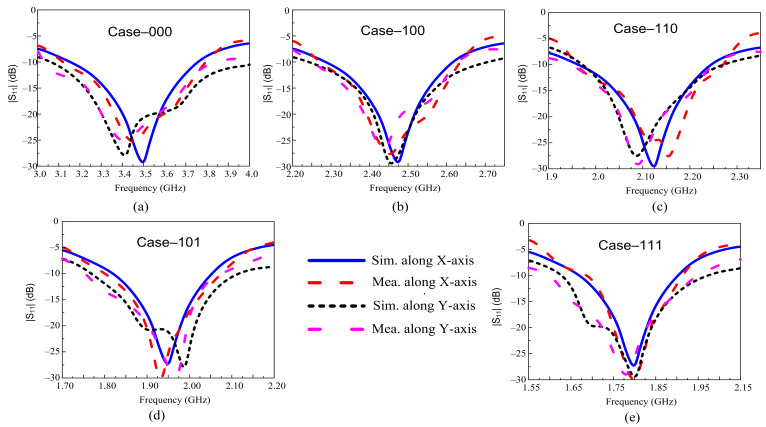
Simulated and measured |S_11_| of the proposed antenna under conformal conditions. (**a**) Case-000; (**b**) Case-100; (**c**) Case-110; (**d**) Case-101; (**e**) Case-111.

**Figure 12 sensors-22-05558-f012:**
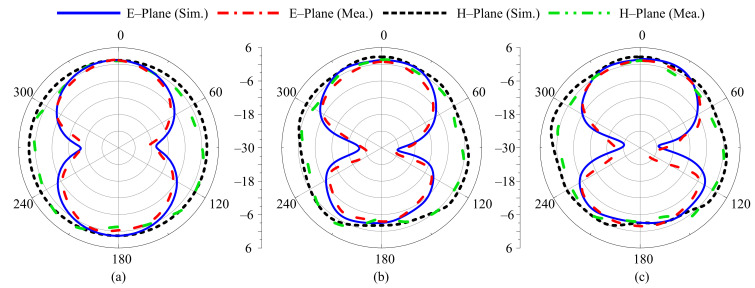
Simulated and measured radiation pattern of proposed antenna at 3.5 GHz for Case-000. (**a**) Flat condition. (**b**) Bend along *x*-axis. (**c**) Bend along *y*-axis.

**Figure 13 sensors-22-05558-f013:**
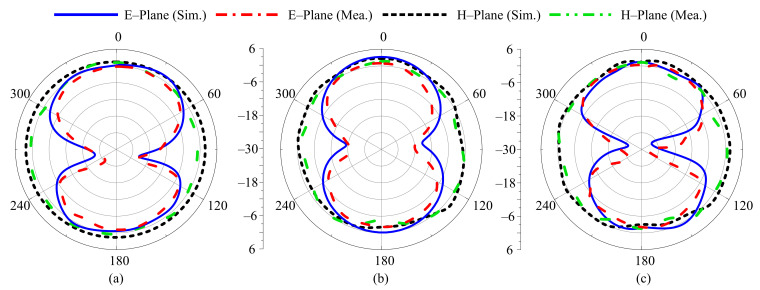
Simulated and measured radiation pattern of proposed antenna at 2.45 GHz for Case-100. (**a**) Flat condition. (**b**) Bend along *x*-axis. (**c**) Bend along *y*-axis.

**Figure 14 sensors-22-05558-f014:**
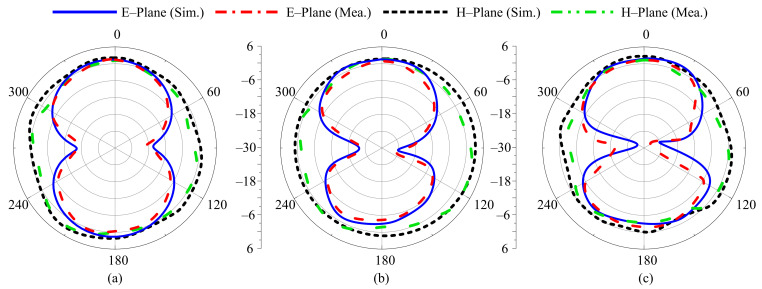
Simulated and measured radiation pattern of proposed antenna at 2.1 GHz for Case-110. (**a**) Flat condition. (**b**) Bend along *x*-axis. (**c**) Bend along *y*-axis.

**Figure 15 sensors-22-05558-f015:**
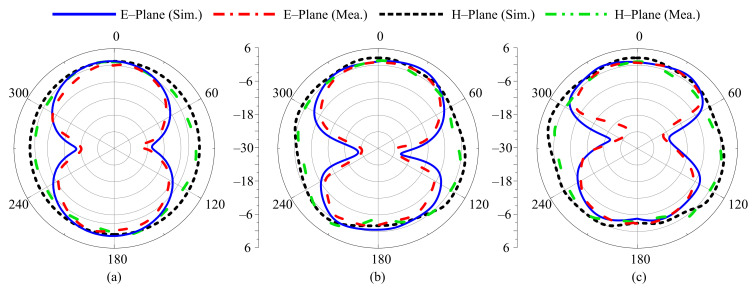
Simulated and measured radiation pattern of proposed antenna at 1.9 GHz for Case-101. (**a**) Flat condition. (**b**) Bend along *x*-axis. (**c**) Bend along *y*-axis.

**Figure 16 sensors-22-05558-f016:**
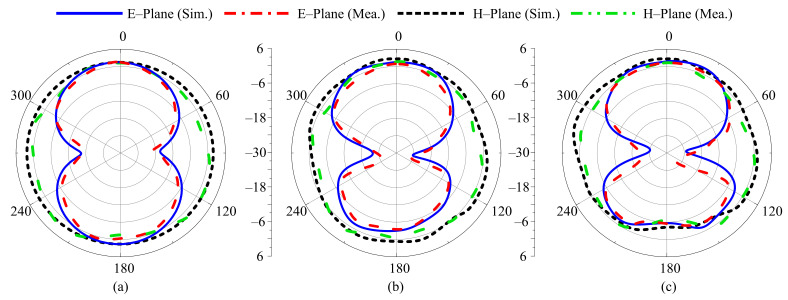
Simulated and measured radiation pattern of proposed antenna at 1.8 GHz for Case-111. (**a**) Flat condition. (**b**) Bend along *x*-axis. (**c**) Bend along *y*-axis.

**Figure 17 sensors-22-05558-f017:**
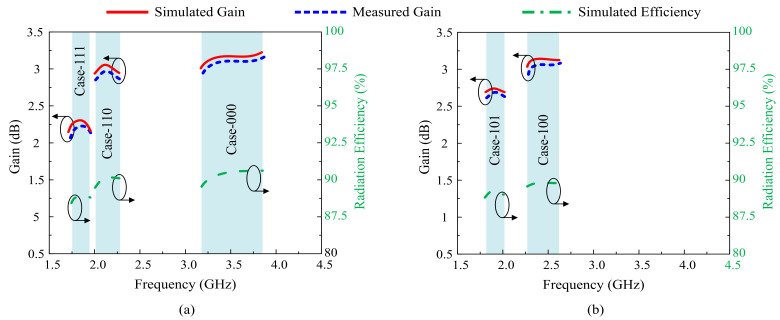
Gain and radiation efficiency of the antenna for the various switching states. (**a**) Case 00, Case 110, and Case 111. (**b**) Case 100 and Case 101.

**Table 1 sensors-22-05558-t001:** Comparison of proposed filtenna with state-of-the-art work for similar applications.

Ref.	Size (mm^3^)	Resonating Frequency (GHz)	Operational Bandwidth (GHz)	Gain (dBi)	Flexible	Reconfigurability	Harmonic Suppression
[15]	70 × 70 × 6.6	5.2	4.85–5.65	2.9	✓	✗	✗
[17]	110 × 88 × 0.3	0.8 2.45 4.0	0.74–0.81 2.4–2.5 3.9–4.1	-	✓	✓	✗
[18]	89 × 83 × 1.5	2.45/3.3	2.35–2.52 3.28–3.38	2.6 0.6	✓	✓	✗
[19]	60 × 60 × 5.1	2.3 2.4 2.5 2.6 2.7	2.27–2.33 2.37–2.44 2.47–2.54 2.56–2.65 2.66–2.75	2.9 3 3.1 3.3 3.4	✓	✓	✗
[20]	59 × 31 × 0.1	2.36 3.64	2.27–2.45 3.5–3.77	0.7 0.79	✓	✓	✗
[21]	20 × 45 × 1.5	2.4 3.5 5.8 9.5	2.35–2.45 3.25–3.7 5.2–5.9 9.2–9.7	4.9 5.6 8.8 10	✓	✓	✗
[22]	24 × 19 × 1.5	2.4 3.8 5.6	2.41–2.52 3.79–3.98 5.46–5.74	2.3 3.3 3.96	✓	✓	✗
[23]	80 × 70 × 1.2	2.4	2.2–2.6	3.3	✗	✗	✓
[24]	78 × 50 × 0.51	2.45	2.3–2.55	5	✗	✗	✓
[25]	60 × 26 × 1.8	2.45	1.99–2.66	-	✗	✗	✓
[26]	40 × 12 × 1.6	2.45	2.19–2.78	2.28	✗	✗	✓
[27]	72 × 72 × 3	3.7 4.8	3.24–4.03 4.44–5.77	6.8 8	✗	✓	✓
[28]	40 × 55 × 0.25	1.4 3	1.375–1.425 2.75–3.25	-	✓	✓	✓
This Work	32 × 25 × 0.25	1.8 1.9 2.1 2.45 3.5	1.66–1.93 1.81–2.05 2.02–2.29 2.27–2.64 3.17–3.82	2.34 2.74 3.02 3.12 3.2	✓	✓	✓

## Data Availability

Not applicable.

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
