# Peer review of "A Frequency-Reconfigurable Filtenna for GSM, 4G-LTE, ISM, and 5G-Sub 6 GHz Band Applications"

_sensors, 2022, doi:10.3390/s22155558_

Round 1
Reviewer 1 Report
1. I don't see reference 6 in your text
(6. Sung, Y. Simple patch antenna with filtering function using two U-slots. J. Electromagn. Eng. Sci. 2021, 21, 425–429).
2. the soldering of our antenna was not professional (you can do a better soldering than that) because the bad soldering has an impact on the quality of the signals and the radiation pattern 2
Author Response
Dear Reviewer,
The authors are very thankful to the reviewer for the detailed comments. We have addressed all the comments and have modified the manuscript. The point to point response to each comment can be found in the attached file.
Hopefully the revision will satisfy the reviewer.
Best Regards
Authors

Reviewer 2 Report
This manuscript proposes a flexible antenna with frequency reconfigurability and harmonic suppression. Overall, there is a large potential for improvement in the present work, both in writing and in antenna performance.
Of the things this reviewer could notice as places for possible improvements (should such be recommended), we could list the following moments:
1. The authors have used diodes to achieve reconfigurability, but the models of the diodes are not provided in the manuscript.
2. Furthermore, it can be observed in Figure 1(b) that the anodes of the three diodes are grounded simultaneously, representing only one state. However, in the different diode bias states not all grounding pads shown in the figure should be grounded, please depict the reconfigurable state in a more exact way.
3. How did the authors represent the diode bias states in the simulation, and is there an equivalent circuit to replace the diode since different PIN diodes correspond to different lumped models?
4. In equation (1), a formula for calculating the effective radius is given according to reference [31], but the reviewer did not find a corresponding equation in [31]. Can (1) be inferred from [31]? If so, please give brief derivation. Besides, please provide the meaning of x0 in equation (3).
5. In Figure 6, the current distribution for the various switching states is given. The antenna configuration shown in Figure 6 is different from that in Figure 1(a) (inverted arrow patch), could this be representative of the proposed antenna?
6. In Figure 8, the simulated and measured |S11| of the antenna is given. However, it cannot show that the antenna has harmonic suppression. It is also necessary to show the radiation pattern of the antenna for various switch state. Please give the simulated and measured radiation results for various switch state.
7. In Section 3.3, the performance of the proposed antenna is investigated under circular foam. For wearables, further research of S-parameters, radiation patterns, and SAR on flat and curved human tissue phantom should be carried out.
8. Some imperfections in the sentences and description:
-p. 2, l. 70: “this work presents is design…”
-p. 3, fig. 2,
p. 8, fig. 8,
p. 9, fig. 10: “S|11|” > |S11|
-p. 5, l. 123: “an additional resonaof nce…” > an additional resonance of
In Table 1, the corresponding information could not be found in reference [13].
In conclusion, to make this work ready for publication, the authors need to make the above improvements.
Thanks.
Author Response

(The authors gave the same response as above.)

Reviewer 3 Report
The paper is very clear and complete. Thank you!
I will begin with the strong points of the paper:
1) The introduction, the revision of the state of the art and the comparison with other designs (Table 1) is very complete and shown in a very clear way. In this sense, the references are really complete.
2) The antenna design covers a demand in the market, and it is appreciated to work on this kind of new antenna designs.
3) The design method is very clear and well explained. Congratulations because it is not easy to find this kind of papers.
4) The results are supported with simulations and measurements of a prototype.
And, I would recommend some improvements before publication and for a future work.
Before publication, I would include some measurements results for far field in the other bands. The authors shown the radiation pattern, gain and efficiency at 3.5 GHz. However, it would be good to show some results in the other working bands of the antenna. Maybe a table with gain and radiation efficiency at 1.8, 1.9, 2.1, 2.45 and 3.5 GHz would be good. Also, some of the radiation patterns, for instance at 1.8 GHz.
For future work, it would be interested to analyze the performance of the antenna when it is bended. The authors could analyze different cases to study the sensitivity of this design.
Author Response

(The authors gave the same response as above.)

Reviewer 4 Report
There is a typing error in row 123, please correct it.
For each case of surface current distribution in Figure6, please provide information for the frequency at which the current distribution is plotted.
In Figure2, Figure8 from (a) to (e), and Figure10 from (a) to (e) please correct the label on the y-axis for all presented graphs.
In Section 3, row 220, the authors pointed out “…(bending along X- and Y-axis).” For a correct presentation, please on Figures 1, 9(a) and 9(b) display the Cartesian coordinate system.
Author Response

(The authors gave the same response as above.)

Round 2
Reviewer 2 Report
Please give the figures regarding to gain and efficiency vs frequency. Thanks.
Author Response
Reviewer 2
Please give the figures regarding to gain and efficiency vs frequency. Thanks
Response: Authors are very thankful to the reviewer for constructive comments. The authors have added the gain and efficiency vs frequency plot in the revised manuscript.
Author Action:
According to the reviewer’s comment, the Gain and Radiation efficiency plots (Figure 17) have been added to the revised manuscript.
